# Screening of the Pandemic Response Box Reveals an Association between Antifungal Effects of MMV1593537 and the Cell Wall of *Cryptococcus neoformans*, *Cryptococcus deuterogattii*, and *Candida auris*

Haroldo C. de Oliveira,[a] Rafael F. Castelli,[a,b] Flavia C. G. Reis,[a,c] Kirandeep Samby,[d] Joshua D. Nosanchuk,[e] Lysangela R. Alves,[a] Marcio L. Rodrigues[a,f]

[a]Instituto Carlos Chagas, Fundação Oswaldo Cruz (Fiocruz), Curitiba, Brazil
[b]Programa de Pós-Graduação em Biologia Parasitária, Instituto Oswaldo Cruz, Fiocruz, Rio de Janeiro, Brazil
[c]Centro de Desenvolvimento Tecnológico em Saúde (CDTS), Fundação Oswaldo Cruz, Rio de Janeiro, Brazil
[d]Medicines for Malaria Venture, Geneva, Switzerland
[e]Department of Microbiology and Immunology and Division of Infectious Diseases, Albert Einstein College of Medicine of Yeshiva University, New York, New York, USA
[f]Instituto de Microbiologia Paulo de Góes, Universidade Federal do Rio de Janeiro, Rio de Janeiro, Brazil

**ABSTRACT** There is an urgent unmet need for novel antifungals. In this study, we searched for novel antifungal activities in the Pandemic Response Box, a collection of 400 structurally diverse compounds in various phases of drug discovery. We identified five molecules which could control the growth of *Cryptococcus neoformans*, *Cryptococcus deuterogattii*, and the emerging global threat *Candida auris*. After eliminating compounds which demonstrated paradoxical antifungal effects or toxicity to mammalian macrophages, we selected compound MMV1593537 as a nontoxic, fungicidal molecule for further characterization of antifungal activity. Scanning electron microscopy revealed that MMV1593537 affected cellular division in all three pathogens. In *Cryptococcus*, MMV1593537 caused a reduction in capsular dimensions. Treatment with MMV1593537 resulted in increased detection of cell wall chitooligomers in these three species. Since chitooligomers are products of the enzymatic hydrolysis of chitin, we investigated whether surface chitinase activity was altered in response to MMV1593537 exposure. We observed peaks of enzyme activity in *C. neoformans* and *C. deuterogattii* in response to MMV1593537. We did not detect any surface chitinase activity in *C. auris*. Our results suggest that MMV1593537 is a promising, nontoxic fungicide whose mechanism of action, at least in *Cryptococcus* spp, requires chitinase-mediated hydrolysis of chitin.

**IMPORTANCE** The development of novel antifungals is a matter of urgency. In this study, we evaluated antifungal activities in a collection of 400 molecules, using highly lethal fungal pathogens as targets. One of these molecules, namely, MMV1593537, was not toxic to host cells and controlled the growth of isolates of *Cryptococcus neoformans*, *C. deuterogattii*, *C. gattii*, *Candida auris*, *C. albicans*, *C. parapsilosis*, and *C. krusei*. We tested the mechanisms of antifungal action of MMV1593537 in the *Cryptococcus* and *C. auris* models and concluded that the compound affects the cell wall, a structure which is essential for fungal life. At least in *Cryptococcus*, this effect involved chitinase, an enzyme which is required for remodeling the cell wall. Our results suggest that MMV1593537 is a candidate for future antifungal development.

**KEYWORDS** antifungals, *Candida auris*, *Cryptococcus*, drug screening

Address correspondence to Marcio L. Rodrigues, marcio.rodrigues@fiocruz.br.
The authors declare no conflict of interest.

The standard process of drug development is costly, time-consuming, and has a low success rate (1). These obstacles are more critical in the case of neglected diseases, which affect populations with health conditions that usually pose serious risks and who

generally cannot afford to buy medicines (2). Faster and more affordable approaches are necessary to fight infectious diseases in general and, more specifically, to fight neglected diseases.

Fungal diseases are associated with high rates of morbidity and mortality (3). The more than 1 million deaths caused each year by fungi are a great cause of concern (4). In the past 2 years, a new major risk for developing fungal diseases has arisen during the Coronavirus disease 2019 (COVID-19) pandemic. *Candida auris*, an emerging global health threat, has been delineated as an increasingly consequential cause of significant nosocomial infections, emphasizing the hazard of *C. auris* to COVID-19 patients, particularly those in intensive care units (5). Pulmonary aspergillosis and cryptococcosis have been reported as complications of COVID-19 (6). Both immunocompromised and immunocompetent patients with COVID-19 are under serious risk of acquiring mucormycosis, which presents a complex clinical challenge (7). In addition, heretofore rare mycoses, such as fungemia caused by *Candida blankii* (8), are also associated with COVID-19 infections.

In contrast to the impacts of fungi on human health, the tools for fighting fungal infections are few, and those already available have shown problems with toxicity, drug resistance, and high costs (3). The urgent need for novel antifungals has accelerated the development of new drugs and/or new formulations of existing drugs. Examples of these initiatives are olorofim, fosmanogepix, rezafungin, oteseconazole, encochleated amphotericin B, nikkomycin Z, and ATI-2307, which are all in the clinical stage of development (9). Ibrexafungerp is first new agent since the approval of echinocandins (10). Other alternatives, including drug repurposing and the screening of compound collections, have shown promising results (11–13).

The Medicines for Malaria Venture (MMV) and the Drugs for Neglected Diseases initiative, in association with industry and academia, have assembled a compound collection called the "Pandemic Response Box" to foster new research into treatments for infectious diseases. The box corresponds to a collection of 400 structurally diverse compounds for screening against infectious diseases and other neglected maladies. Within this compound collection, inhibitors of the replication of SARS Cov 2 (14) and Zika virus (15), in addition to anti-amoebic drugs (16), antimalarials (17), antischistosomals (18), and inhibitors of *Mycobacterium abscessus* (19) have been described. In this study, we screened the Pandemic Response Box collection for activity against three fungal pathogens with high lethality and exceedingly ineffective treatment options: *Cryptococcus neoformans*, *Cryptococcus deuterogattii*, and *C. auris*. Five compounds showed promising activity, but one of them, MMV1593537, killed all three pathogens and displayed low toxicity to mammalian cells. MMV1593537 has been demonstrated to have antibacterial activity against *Acinetobacter baumannii* (20), but its activity against *Cryptococcus* and *C. auris* has not been reported. We observed that MMV1593537 affects the structure of cell walls in *Cryptococcus* spp. and *C. auris*. In *C. neoformans* and *C. deuterogattii*, these alterations were accompanied by the induction of surface-associated chitinase activity. Our results indicate that MMV1593537 is a promising antifungal which affects the cell walls of major fungal pathogens.

## RESULTS

**Identification of compounds with antifungal activity against *Cryptococcus* spp. and *C. auris*.** In our screen, 23 out of the 400 compounds inhibited at least 50% of the growth of *C. neoformans* H99 at 10 $\mu$M. Out of these 23 compounds, 6 inhibited more than 90% of fungal growth. When we tested the 400 compounds against *C. deuterogattii* R265, 81 molecules inhibited at least 50% of the fungal growth, with 21 of these compounds inhibiting more than 90% of growth. Of the 400 compounds, 24 inhibited at least 50% of the growth of both *C. auris* strains, and 16 inhibited more than 90%. These results are illustrated in Fig. 1.

Five compounds were identified as promising antifungal agents in our screen against the four *Cryptococcus* spp. and *C. auris* strains (Table 1). MMV1633966 inhibited approximately 90% of the growth of all strains. MMV019724 inhibited only 68% of the growth of *C. neoformans* H99, but its inhibitory activity against the other strains was in the 90%

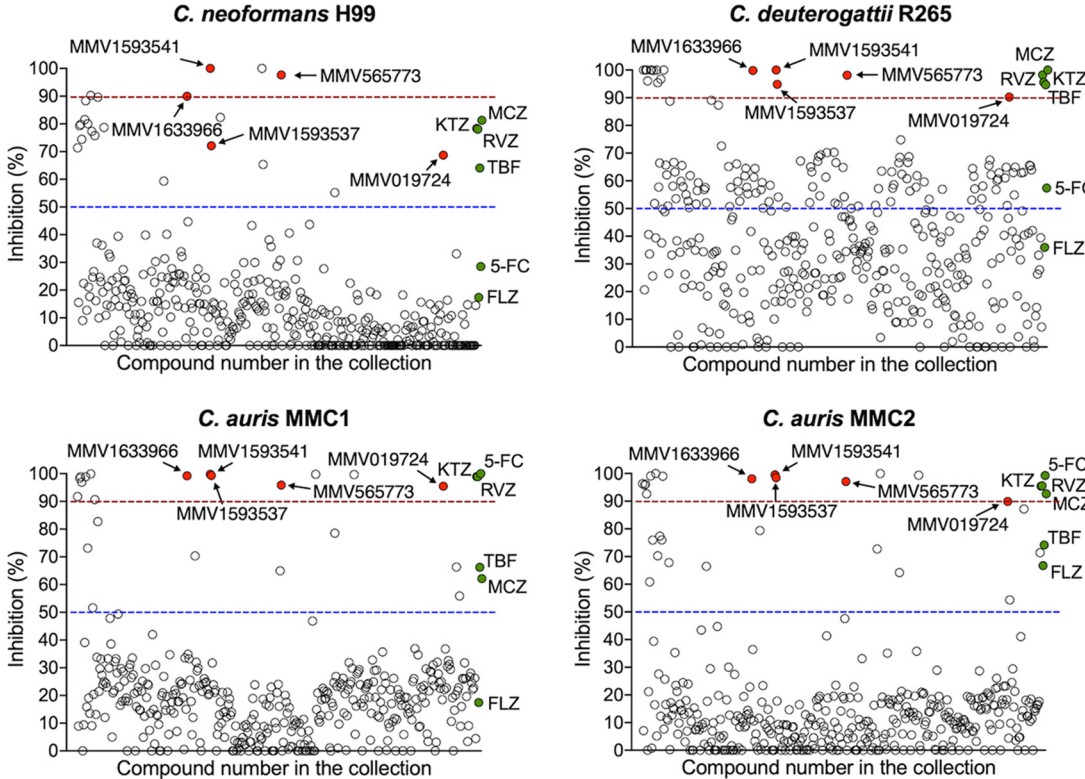

**FIG 1** Antifungal activities against *C. neoformans* H99, *C. deuterogattii* R265, or *C. auris* MMC1 and MMC2 in the Pandemic Response Box collection. Activities of known antifungals are represented by green circles (5-FC, flucytosine; RVZ, ravuconazole; MCZ, miconazole; TBF, terbinafine; FLZ, fluconazole). Most of the compounds had activities below the 50% growth inhibition cutoff. We selected five of the most active compounds for further tests based on their ability to inhibit the growth of *C. neoformans*, *C. deuterogattii*, and *C. auris* in the 70 to 100% range. These compounds are indicated by red circles.

range. The most effective antifungal activity was observed for the compound MMV1593541, which inhibited 100% of the growth of both *Cryptococcus* strains and 99% of that of both *C. auris* strains. MMV565773 also showed potent antifungal activity, with inhibition levels higher than 90% for all strains. The compound MMV1593537 was less effective against *C. neoformans* H99 (72% of growth inhibited), although it showed more than 90% growth inhibition for *C. deuterogattii* R265 and both *C. auris* strains.

**Antifungal susceptibility testing of the selected compounds.** We determined the minimum inhibitory concentrations (MICs) and minimum fungicidal concentrations (MFCs) of MMV1633966, MMV019724, MMV1593541, MMV565773, and MMV1593537 for the cryptococcal and *C. auris* strains using the recommendations of EUCAST protocol (21).

MMV1633966 almost completely inhibited the growth of *C. auris* (MMC1 and MMC2 strains) and *C. deuterogattii* R265 at 2.5 $\mu$M, and similar results at 5 $\mu$M were observed for *C. neoformans* H99 (Fig. 2A). At higher concentrations, MMV1633966 produced controversial results. Unexpectedly, in the dose-response tests, the percentage of growth inhibition at 10 $\mu$M was repeatedly lower than that observed in the full screenings. In addition, fungal growth was less affected at >10 $\mu$M, with the cryptococcal strains demonstrating no significant effects at 20 $\mu$M. The MIC of MMV019724 also corresponded to 2.5 $\mu$M for *C. auris* MMC1 and 5 $\mu$M for the other strains (Fig. 2B). MMV1593541 had a MIC of 10 $\mu$M for all four strains (Fig. 2C). MMV565773 showed MICs of 10 $\mu$M for *C. auris* MMC1 and MMC2, and 5 and 2.5 $\mu$M for *C. neoformans* H99 and *C. deuterogattii* R265, respectively (Fig. 2D). Finally, the MIC of MMV1593537 against both *C. auris* and the *Cryptococcus* spp. strains was 5 $\mu$M (Fig. 2E).

We also measured the ability of the five compounds to kill *C. auris*, *C. neoformans*, and *C. deuterogattii* (Fig. 3). Fungicidal activities were observed for MMV1593541 (*C. auris* MMC1, 20 $\mu$M; *C. auris* MMC2, 10 $\mu$M; *C. neoformans* H99, 20 $\mu$M; *C. deuterogattii* R265,

**TABLE 1** Information on the compounds selected from the MMV Pandemic Response Box collection[a]

| Compound | Structure[b] | Original reported activity | Growth inhibition at 10 $\mu$M (%) | | | |
|---|---|---|---|---|---|---|
| | | | Cryptococcus | | C. auris | |
| | | | H99 | R265 | MMC1 | MMC2 |
| MMV1633966 | | Antibacterial | 89 | 99 | 99 | 98 |
| MMV019724 | | Antibacterial | 68 | 90 | 95 | 89 |
| MMV1593541 | | Antibacterial | 100 | 100 | 99 | 99 |
| MMV565773 | | Antibacterial | 97 | 98 | 95 | 98 |
| MMV1593537 | | Antibacterial | 72 | 94 | 99 | 98 |

[a]MMV, Medicines for Malaria Venture.
[b]Structures were obtained from ChEMBL (https://www.ebi.ac.uk/chembl/), a database of bioactive molecules with drug-like properties.

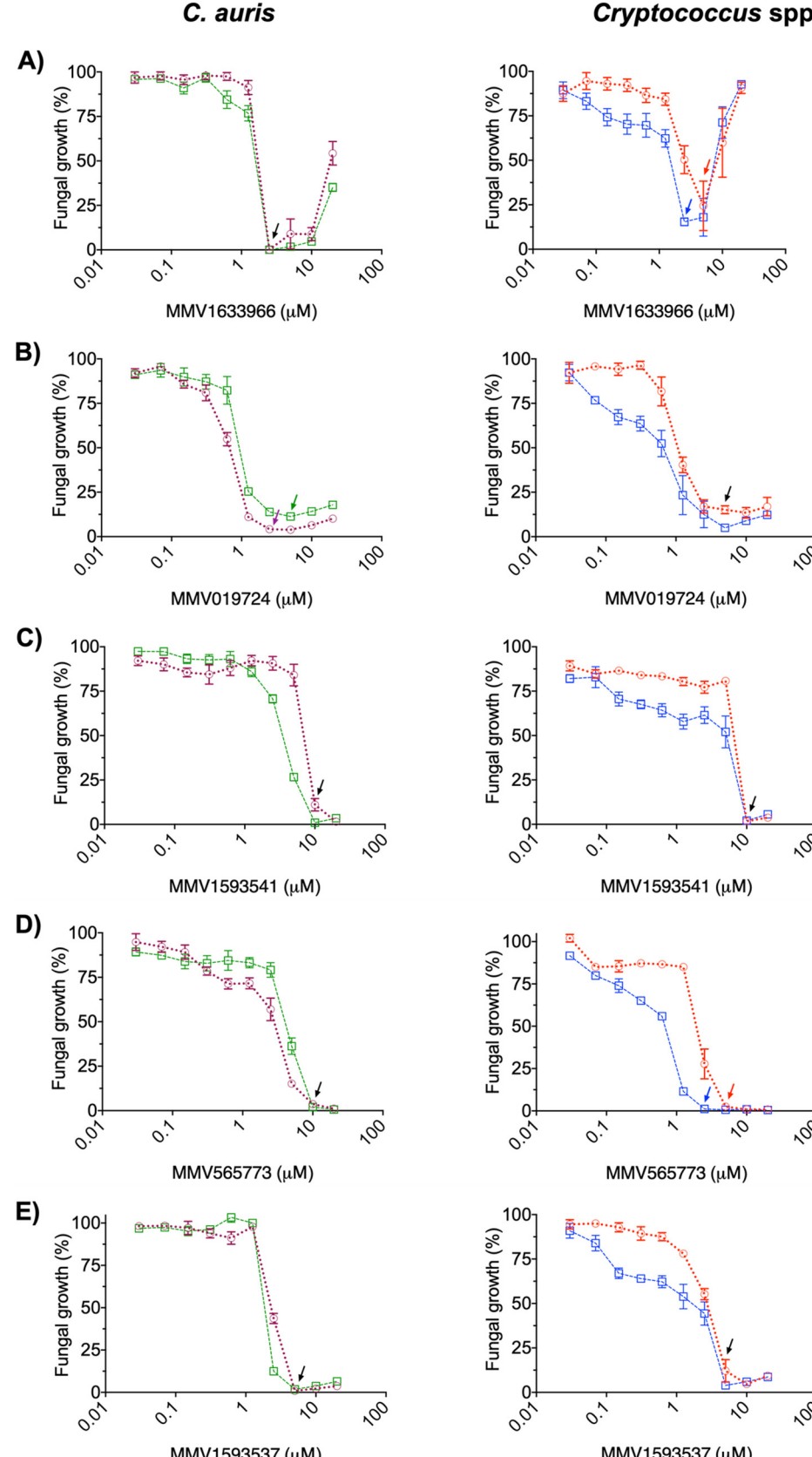

**FIG 2** Determination of MICs against different isolates of *Cryptococcus* and *C. auris* using MMV1633966 (A), MMV019724 (B), MMV1593541 (C), MMV565773 (D), and MMV1593537 (E). In the tests with *C. auris*, purple

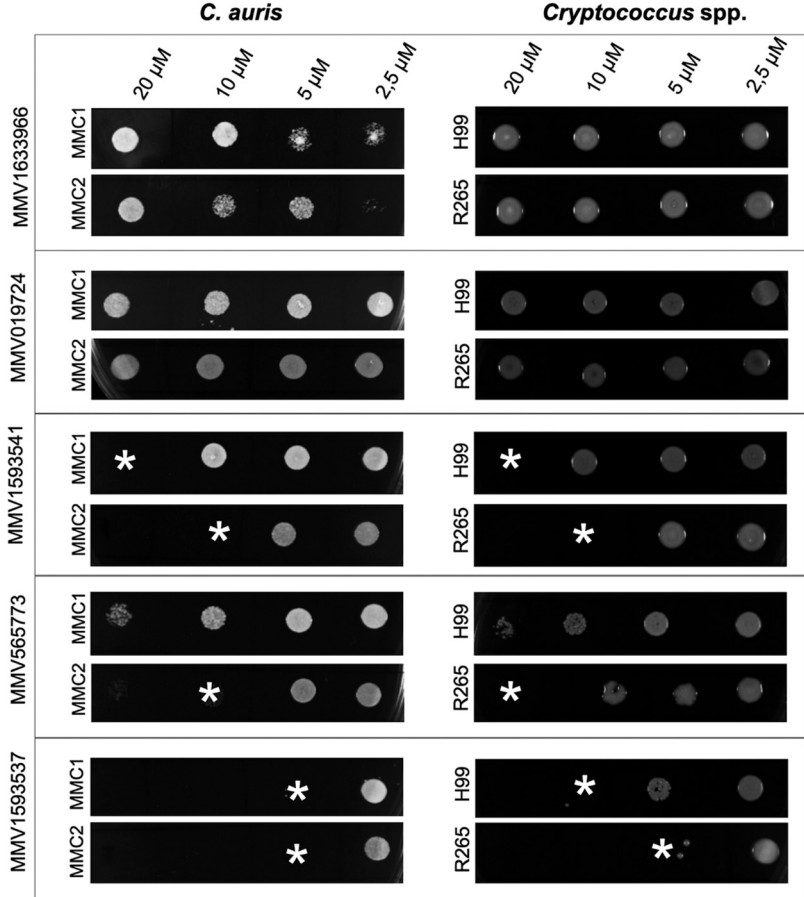

**FIG 3** Minimum fungicidal concentration (MFC) determination for MMV1633966, MMV019724, MMV1593541, MMV565773, and MMV1593537 against *Cryptococcus* and *C. auris* strains. MFC was defined as the minimum concentration at which no fungal growth was observed (indicated by asterisks). MMV1633966 and MMV019724 did not result in fungicidal activity.

10 $\mu$M), MMV565773 (*C. auris* MMC2, 20 $\mu$M; *C. deuterogattii* R265, 20 $\mu$M), and MMV1593537 (*C. auris* MMC1, 5 $\mu$M; *C. auris* MMC2, 5 $\mu$M; *C. neoformans* H99, 10 $\mu$M; *C. deuterogattii* R265, 5 $\mu$M). MMV1633966 and MMV019724 showed no fungicidal activity, while MMV565773 was unable to kill *C. auris* MMC1 and *C. neoformans* H99.

**Cytotoxicity of the selected compounds.** Cytotoxicity was the final filter we applied in the selection of our compounds for cell biology and biochemical tests. We tested the effects of the five selected compounds at 1 to 10 $\mu$M on the viability of cultured macrophages (RAW 264.7), based on the roles of these cells in the control and/or dissemination of cryptococci (22). At their MICs, MMV1633966 and MMV1593537 did not display any significant toxicity (Fig. 4A and B). MMV019724 showed significant toxicity, killing 79% of the cells at its MIC (Fig. 4C). At their MICs, MMV1593541 and MMV565773 killed 58% and 35% of macrophages, respectively (Fig. 4D and E).

The antifungal and toxicity profiles of the 5 compounds are summarized in Table 2. Considering that MMV1593537 was fungicidal against all tested strains and showed negligible toxicity, this compound was selected for our further tests.

**Antifungal characterization of MMV1593537.** We tested whether MMV1593537 would be active against other isolates belonging to the *Cryptococcus* and *Candida* genera. To

**FIG 2** Legend (Continued)

lines represent strain MMC1 and green lines represent strain MMC2. In the tests with *Cryptococcus*, red lines represent *C. neoformans* H99 and blue lines represent *C. deuterogattii* R265. Minimum inhibitory concentrations (MICs) are indicated by arrows.

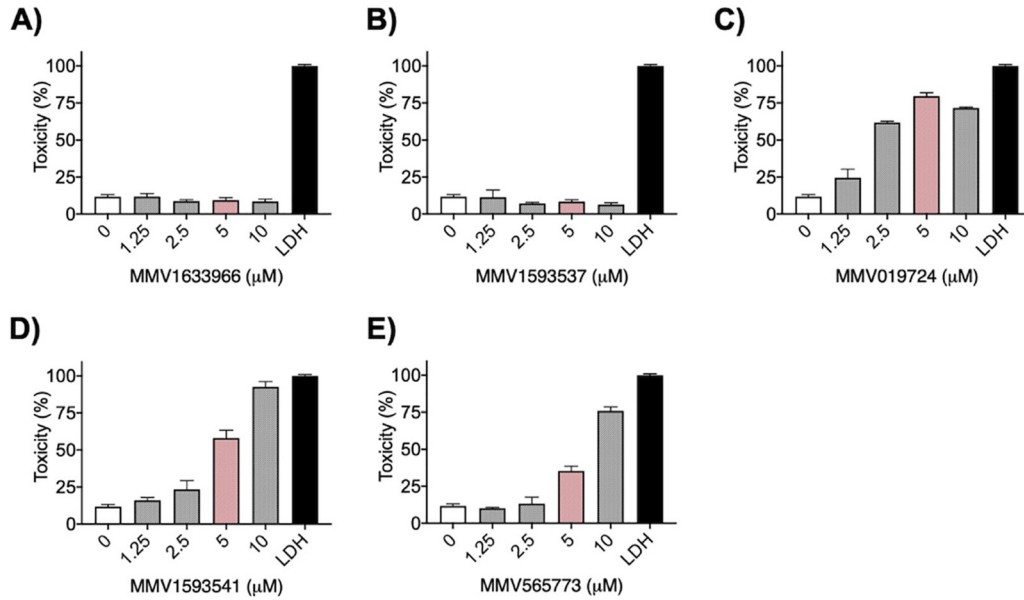

**FIG 4** Dose-dependent toxicity profiles of MMV1633966 (A), MMV1593537 (B), MMV019724 (C), MMV1593541 (D), and MMV565773 (E) and against RAW 264.7 macrophages. Lactate dehydrogenase (LDH) represents the control of cell death. Black bars represent death controls. White bars represent the absence of antifungal compounds. Pink bars represent compound MICs, while gray bars represent additional concentrations tested for toxicity.

address this question, we tested MMV1593537 against isolates of *Cryptococcus neoformans* (*n* = 9), *C. deuterogattii* (*n* = 6), *C. gattii* (*n* = 2), *Candida auris* (*n* = 8), *C. albicans* (*n* = 2), *C. parapsilosis* (*n* = 1), and *C. krusei* (*n* = 1). MMV1593537 was active against all of them, with MICs in the ranges of 2.5 to 5 μM for *C. neoformans*, 0.625 to 5 μM for *C. deuterogattii*, 5 μM for *C. gattii*, 5 to 10 μM for *C. auris* and *C. albicans*, and 10 μM for *C. krusei*. These results (Table 3) suggest a large spectrum of antifungal activity for MMV1593537.

We performed fungal growth curves at different concentrations of MMV1593537 (Fig. 5) to select subinhibitory concentrations for the evaluation of the cellular and biochemical effects of this compound. For all strains, the concentration chosen was 2.5 μM. At this concentration, *C. neoformans* H99, *C. deuterogattii* R265, and *C. auris* MMC2 could grow, but at reduced rates in comparison with those in the control systems (no compound). This concentration was selected for subsequent experiments with these three strains based on the combination of antifungal effect (reduced growth rates) with the maintained replication ability of the fungi. For *C. auris* MMC1, no intermediary effects were observed: the cells grew normally at 0 to 2.5 μM MMV1593537, whereas their growth was completely inhibited at 5 μM. Therefore, for this strain, we selected 2.5 μM because this was 0.5 MIC, which is a common concentration for sub-inhibitory compound testing, and for consistency with the other strains.

**TABLE 2** Inhibitory and fungicidal properties and toxicity of compounds MMV1633966, MMV019724, MMV1593541, MMV565773, and MMV1593537 (highlighted as the most promising compound)

| | Antifungal activity (MICs and MFCs [μM])[a] | | | | | | | | |
|---|---|---|---|---|---|---|---|---|---|
| | *C. auris* MMC1 | | *C. auris* MMC2 | | *C. neoformans* H99 | | *C. deuterogattii* R265 | | Toxicity at the MIC |
| Compound | MIC | MFC | MIC | MFC | MIC | MFC | MIC | MFC | |
| MMV1633966 | 2.5 | >20 | 2.5 | >20 | 5 | >20 | 2.5 | >20 | No |
| MMV019724 | 2.5 | >20 | 5 | >20 | 5 | >20 | 5 | >20 | Yes |
| MMV1593541 | 10 | 20 | 10 | 10 | 10 | 20 | 10 | 10 | Yes |
| MMV565773 | 10 | >20 | 10 | 10 | 5 | >20 | 2.5 | 20 | Yes |
| MMV1593537 | 5 | 5 | 5 | 5 | 5 | 10 | 5 | 5 | No |

[a]MIC, minimum inhibitory concentration; MFC, minimum fungicidal concentration.

**TABLE 3** MICs of MMV1593537 against multiple isolates of the *Cryptococcus* and *Candida* genera[a]

| Pathogen | Isolate | MIC ($\mu$M) |
|---|---|---|
| *Cryptococcus* | | |
| C. neoformans | H99 | 5 |
| | 3Pb3 | 5 |
| | 17A1 | 5 |
| | 23Pb2 | 5 |
| | 19Pb4 | 2.5 |
| | Cg366 | 5 |
| | Cn161 | 5 |
| | Cn222 | 5 |
| | Cn116 | 5 |
| C. deuterogattii | R265 | 5 |
| | Cg460 | 0.625 |
| | Cg221 | 1.25 |
| | Cg158 | 5 |
| | Cg456 | 2.5 |
| | Cg188 | 2.5 |
| C. gattii | Cg365 | 5 |
| | Cg367 | 5 |
| | | |
| *Candida* | | |
| C. auris | MMC1 | 5 |
| | MMC2 | 5 |
| | CDC383 | 10 |
| | CDC388 | 5 |
| | CDC384 | 10 |
| | CDC390 | 5 |
| | CDC387 | 10 |
| | CDC385 | 5 |
| C. albicans | ATCC 90028 | 10 |
| | ATCC MYA-2876 | 5 |
| C. parapsilosis | ATCC 6258 | 10 |
| C. krusei | ATCC 22019 | 10 |

[a]Cryptococcal (46, 47) and *C. auris* isolates (45) were previously characterized. Isolates of *C. albicans* (ATCC MYA-2876 and ATCC 90028), *C. parapsilosis* (ATCC 6258), and *C. krusei* (ATCC 22019) are available from the American Type Culture Collection (ATCC).

**Effects of MMV1593537 on fungal morphology.** The general aspects of MMV1593537-treated cells were observed by scanning electron microscopy (SEM). In contrast to that in the control (dimethyl sulfoxide [DMSO]-treated) cells, MMV1593537 induced the formation of aggregates in both strains of *C. auris*, with a much more pronounced effect manifested by strain MMC2 (Fig. 6).

The induction of large aggregates was not observed in the *C. neoformans* and *C. deuterogattii* models, although some of the daughter cells seemed to remain attached to their parental cells in the strain H99 of *C. neoformans* (Fig. 7). However, the most pronounced effect of MMV1593537 was the reduction of capsule for both strains. The capsular fibers were smaller and less abundant. The latter supposition was confirmed by measurement of the capsule size. In both strains, capsular dimensions were significantly reduced ($P < 0.05$), with a more pronounced effect occurring in *C. deuterogattii* R265.

Both cell division and capsule formation are events which require intact and fully functional cell walls (23, 24). We then tested whether the formation of aggregates in *C. auris* and the reduced capsules in *Cryptococcus* were indications of cell wall defects. Therefore, we next evaluated the general aspects of the fungal cell wall after exposure to MMV1593537.

Since the cell walls of different fungi vary considerably in their composition (25), we chose chitin, a conserved cell wall component across all species of fungi, as the molecule to analyze in both *C. auris* and *Cryptococcus*. In this analysis, we included the detection of chitooligomers, well-known surface components of the fungal cell wall which derive from chitin (26). In fact, chitooligomers are formed by the enzymatic hydrolysis of chitin during cell division. The cell walls of both *C. auris* and *Cryptococcus* were efficiently stained by Calcofluor

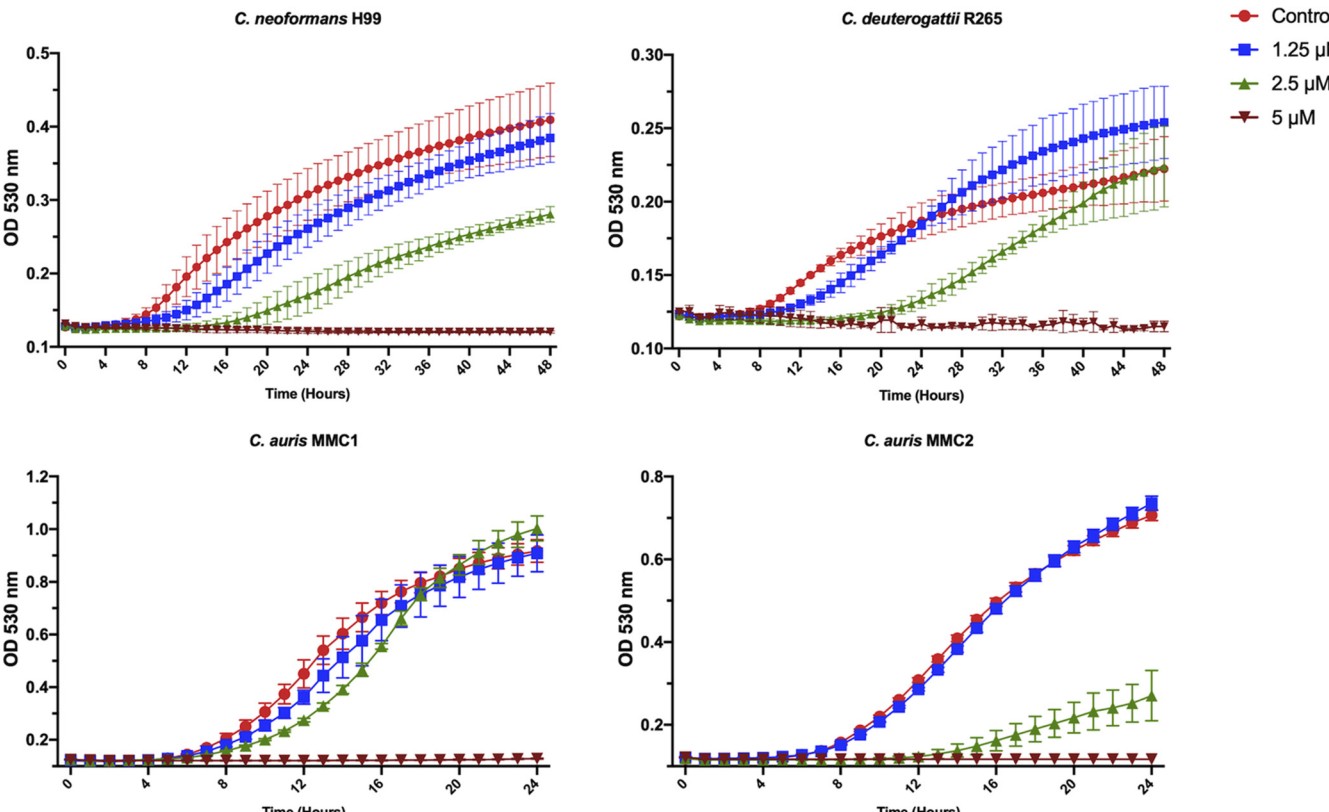

**FIG 5** Growth of *C. neoformans*, *C. deuterogattii*, and *C. auris* in the presence of varying concentrations of MMV1593537.

white (CFW) (Fig. 8). Changes in the pattern of chitin staining comparing control and MMV1593537-treated fungi were not visually clear, which led us to quantify the Calcofluor-derived blue fluorescence in all systems using the Image J software. No significant differences were observed when *Cryptococcus* spp. were tested, and a discrete increase ($<$10%) in fluorescence was observed exclusively in strain MMC2 of *C. auris* after exposure to MMV1593537 (data not shown). These results differed from those obtained when chitooligomers were analyzed. In control cells, both *C. auris* and *Cryptococcus* manifested the typical pattern of WGA staining (27), revealing a distribution of chitooligomers in

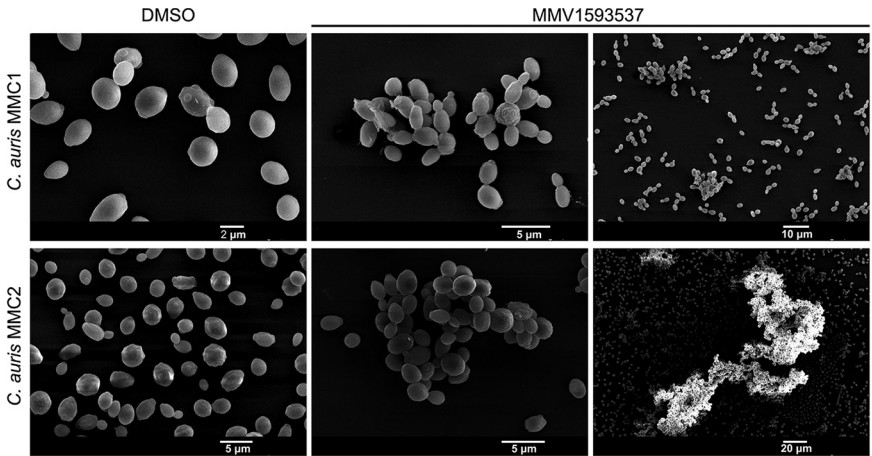

**FIG 6** The effects of dimethyl sulfoxide (DMSO, control) or MMV1593537 on the morphology of MMC1 and MMC2 isolates of *C. auris*. MMV1593537 induced an aggregate phenotype in both isolates. The MMV1593537 panels illustrate similar results at different magnifications.

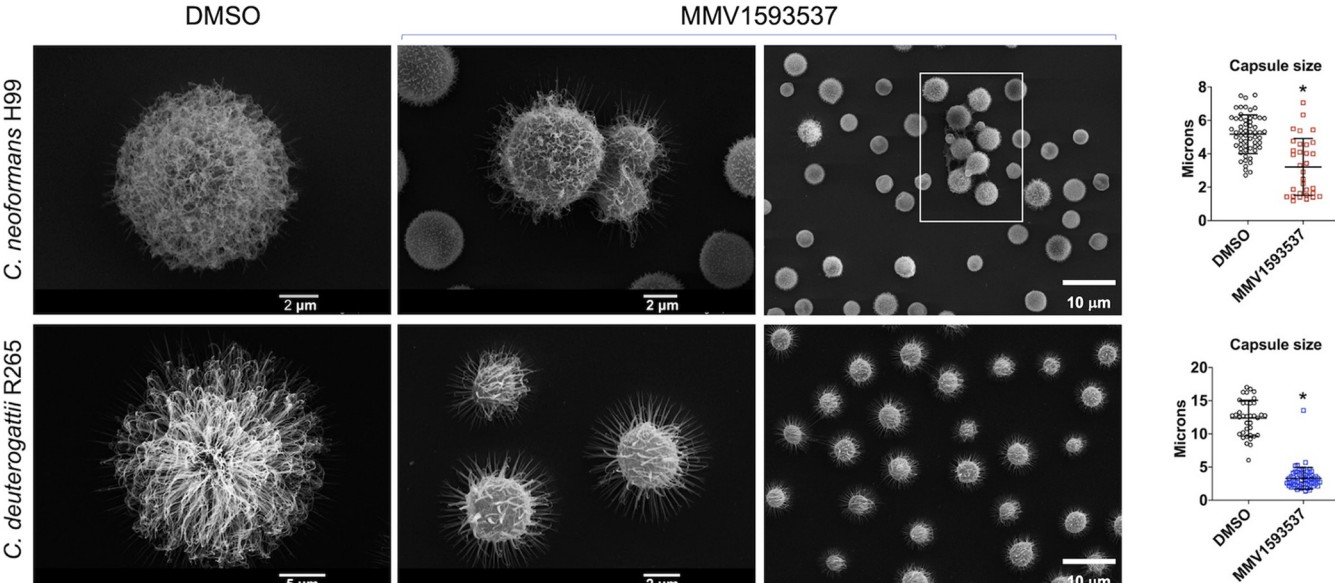

**FIG 7** The effects of DMSO (control) or MMV1593537 on the morphology of cryptococci. The capsular properties (microscopic panels) were apparently affected after treatment with MMV1593537, mostly in *C. deuterogattii* R265. In *C. neoformans*, but not in *C. deuterogattii*, discrete aggregates were observed (boxed area in the lower-magnification panel). Determination of capsular dimensions (right panels) confirmed the visual observation that MMV1593537 induced a reduction in capsule size. Asterisks denote statistical significance ($P < 0.05$, according to analysis of variance with Tukey's *post hoc* test) in comparison with control conditions.

structures which resembled bud scars. These results agreed with previous literature data which showed the same pattern of lectin staining (26). In MMV1593537-treated cells, however, chitooligomers were more abundantly detected. This increased detection of surface chitooligomers has been associated with increased chitinase activity in *C. neoformans* (26–28). Therefore, we examined whether MMV1593537 treatment interfered with chitinase activity in *C. auris* and *Cryptococcus* spp.

**Chitinase activity.** Since the changes in the detection of chitooligomers were observed at the cell wall, we measured the activity of chitinase in intact cells, assuming that the possible hydrolysis of chitin which resulted in chitooligomer abundance was surface-associated. We could not detect any chitinase activity in *C. auris* under the conditions used in our study,

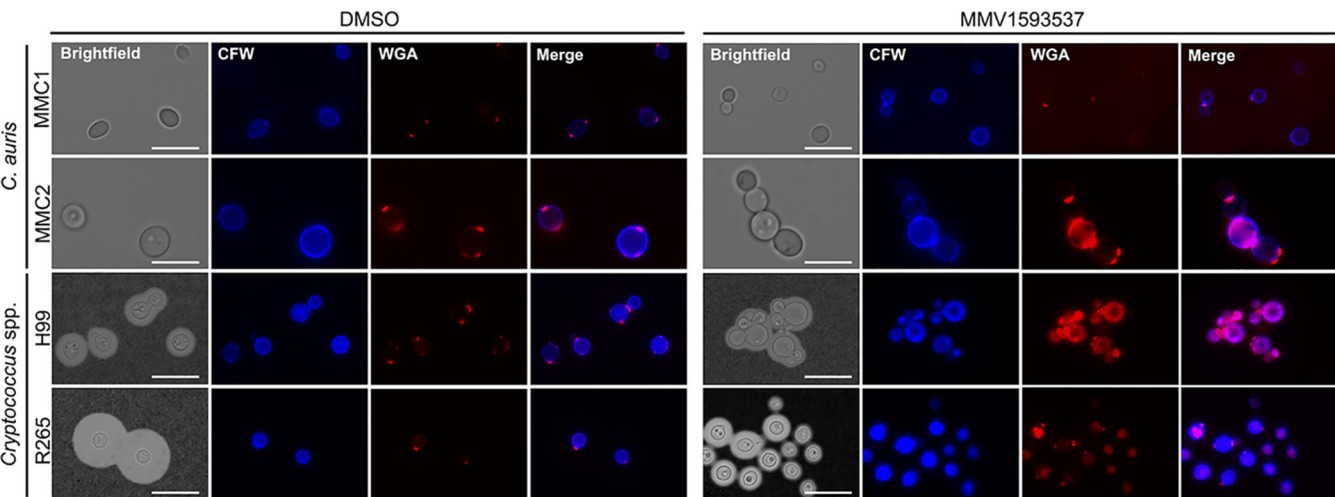

**FIG 8** The effects of MMV1593537 on the cell wall architecture of *C. neoformans*, *C. deuterogattii*, and *C. auris*. In the brightfield panels, India ink-counterstained cryptococci were included for visualization of the capsule, which was not applicable to *C. auris*. In the fluorescence panels, cells were stained for cell wall chitin with Calcofluor white (CFW, blue fluorescence) and for chitooligomers with fluorescent wheat germ agglutinin (WGA, red fluorescence). As demonstrated in the WGA and merge panels, MMV1593537 induced an increase in detectable cell wall chitooligomers. Scale bars represent 10 μm.

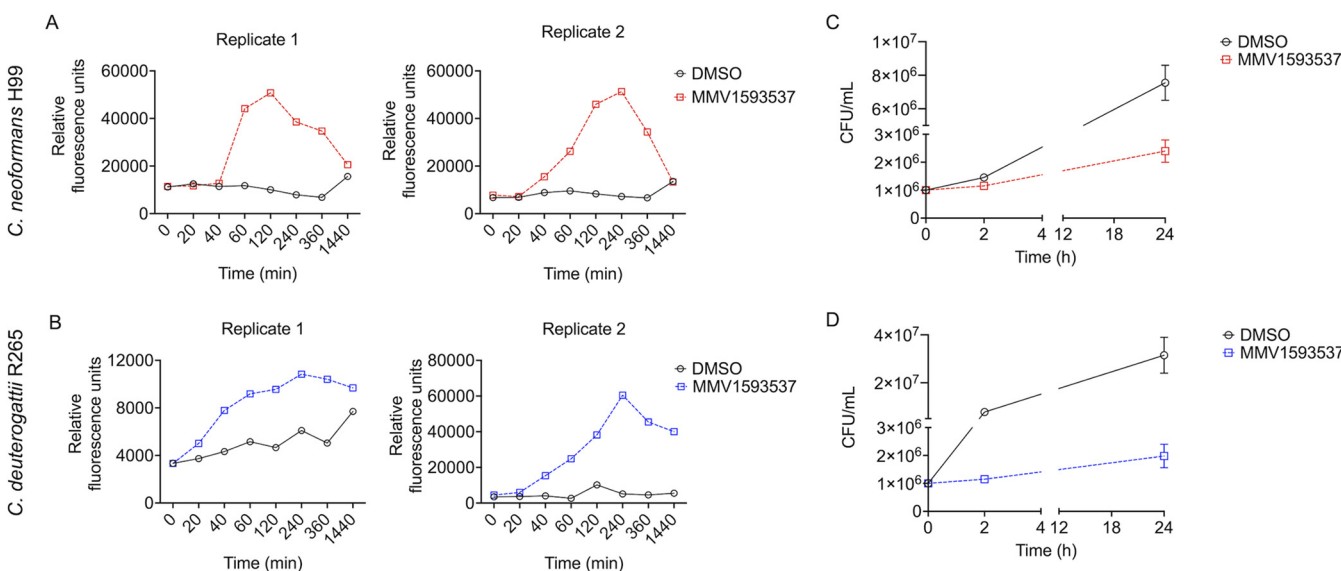

**FIG 9** The effects of MMV1593537 on surface-associated chitinase activity in *C. neoformans* H99 (A) and *C. deuterogattii* R265 (B). Independent replicates are shown. Compared to DMSO (vehicle), MMV1593537 induced peaks of enzyme activity after 120 (*C. neoformans* H99) or 240 (*C. deuterogattii* R265) min of incubation. To check whether the decreased chitinase activity in drug-treated cells after 120 (*C. neoformans* H99) or 240 (*C. deuterogattii* R265) min was a consequence of cell death, we determined CFU at the 0, 2, and 24 h time points for *C. neoformans* H99 (C) and *C. deuterogattii* R265 (D). Fungal viability increased with time in both control (DMSO)- and MMV1593537-treated cells.

even in the control cells (data not shown). However, when we tested the chitinase activity of both control and MMV1593537-treated cryptococci, we observed that this compound induced peaks of enzyme activity after 120 (*C. neoformans* H99) or 240 (*C. deuterogattii* R265) min of incubation with the drug (Fig. 9A and B). In both species, enzyme activity tended to decrease after these periods of incubation. The induction of chitinase peaks by MMV1593537 agreed with results which demonstrated that this compound induced increased detection of chitooligomers, the products of the enzymatic degradation of chitin. We tested whether the decrease in chitinase activity after 120 (*C. neoformans* H99) or 240 (*C. deuterogattii* R265) min of incubation was a result of cell death. To address this question, we determined cell viability by counting CFU at the 0, 2, and 24 h time points (Fig. 9C and D). As expected, in the presence of 2.5 $\mu$M MMV1593537, fungal growth was slower than that in the control conditions (DMSO, no drug). However, there was no evidence of any decrease in cell viability.

## DISCUSSION

Screening of compound collections has emerged as a promising approach for the development of antimicrobials. Several candidates for antifungal development have been recently characterized using this approach. For instance, a collection of approximately 50,000 compounds was screened by Mor et. al. (29) to identify molecules which inhibited the growth of *C. neoformans*. Using this approach, the compounds *N'*-(3-bromo-4-hydroxybenzylidene)-2-methylbenzohydrazide and its derivative, 3-bromo-*N'*-(3-bromo-4-hydroxybenzylidene) benzo-hydrazide were found to inhibit the synthesis of fungal, but not mammalian, glycolipids. The two compounds affected fungal cell morphology and controlled infection *in vivo* (29), which makes them promising candidates for human trials. These results efficiently illustrate how the screening of compound collections can identify antifungal activities and their mechanisms of action, fostering antifungal development.

In our study, we used a much smaller compound collection to identify five candidates to fight *Cryptococcus* and *C. auris*. We applied a combination of go/no go filters to select the most promising compounds within this group. For instance, compound MMV1633966 was disregarded because, despite inhibiting fungal growth at relatively low concentrations, it proved to lack efficacy at higher concentrations. The profile of fungal growth in the presence of MMV1633966 resembled the so-called paradoxical

effect observed for echinocandins, which has been described as the reversal of growth inhibition at high doses of the antifungals (30). This effect has not been observed for the most recently developed antifungals. For instance, ibrexafungerp, a novel oral triterpenoid antifungal, has a concentration-dependent fungicidal activity without paradoxical effects (31). Other compounds (MMV019724, MMV1593541, and MMV565773) showed considerable toxicity to macrophages in our study. However, MMV1593537 demonstrated fungicidal activity against *C. neoformans*, *C. deuterogattii*, and *C. auris* and low toxicity to murine macrophages. Together, these results led us to select MMV1593537 for more detailed analysis of its antifungal effects.

Besides exerting fungicidal effects, MMV1593537 showed additional activities which could be beneficial for treating fungal infections. Of note, all of these activities were cell wall-associated. The cryptococcal capsule is thought to be the most important virulence factor in the *Cryptococcus* genus (24). The assembly of the capsule depends on the proper interaction between capsular and cell wall polysaccharides (32). In our study, MMV1593537 efficiently repressed capsule formation in both *C. neoformans* and *C. deuterogattii*. Considering the role of the capsule during infection (24), capsule reduction could synergize with the killing activity of MMV1593537, favoring the control of cryptococcosis. The observation of reduced capsules following exposure to MMV1593537 suggest that this compound interferes with the cell wall, the anchoring cellular site of the capsule (32). Accordingly, exposure to MMV1593537 demonstrated additional cell-associated activities, affecting cellular separation between mother and daughter cells in *Cryptococcus* and inducing large cellular aggregates in *C. auris*.

In agreement with its possible interference with the cell wall, treatment with MMV1593537 resulted in increased detection of chitooligomers in both *Cryptococcus* spp. and *C. auris*. Chitooligomers are a class of molecules that derive from chitin, a major cell wall polysaccharide in fungi (33). The formation of chitooligomers results from the enzymatic hydrolysis of chitin by chitinases (27), an event that is necessary for cell wall separation during replication (33). For fungal growth, chitin synthesis and hydrolysis are similarly important (33), and an imbalance in this synthesis/degradation regulation can affect replication. Chitinases play diverse roles in cell wall remodeling during different stages of the fungal life cycle, influencing cell wall integrity, cell separation, mating, and stress resistance (34, 35). In *C. neoformans*, stressing conditions induced cell-associated chitinase activity (34).

Chitin is also the target of exogenous enzymes with chitinase activity. In plant and animal hosts, these enzymes have important roles in immunity (36). Mammals do not synthesize chitin, but they do produce two chitinases, chitotriosidase (Chit1) and acidic mammalian chitinase (AMCase), which can hydrolyze chitin (36). Lung AMCase is active in rats with pulmonary cryptococcosis (37, 38), and serum chitotriosidase levels are elevated in response to systemic aspergillosis (39). Increased Chit1 expression is protective in a murine model of cryptococcosis (40), and Chit1 is antifungal both *in vitro* and *in vivo* (41). Similarly, a *Penicillium oxalicum* chitinase showed fungicidal activity against *Sclerotinia sclerotiorum* (42). In plants, chitinase overexpression results in resistance to fungal infections (43). Together, these observations point to an antifungal role of mammalian, plant, and fungal chitinases that could be biochemically related to the activity of MMV1593537. Importantly, we cannot rule out the possibility that other mechanisms are involved in the antifungal activity of MMV1593537, since chitinase activity was not detected in *C. auris*.

Chitinase-related mechanisms of antifungal activity could be similar to those found in the rat model of invasive pulmonary aspergillosis. In these animals, aspergillosis caused a dramatic increase in chitotriosidase and AMCase activity even when rats were treated with caspofungin (44). *In vitro* tests demonstrated that both chitinases combined had a direct effect on the fungus *in vitro*, and they were needed to lyse the fungal cell wall upon caspofungin exposure (44). In that study, caspofungin altered the fungal surface in such a way that the two chitinases, when combined, could lyse the fungal cell wall and assist in clearing the fungal pathogen. In our model, we speculate that MMV1593537 induces peaks of chitinase activity that, based on what was reported

in the literature, could kill cryptococci by affecting the cell wall. Since we have not been able to demonstrate surface chitinase activity in *C. auris*, we still do not know if a similar mechanism could occur in this fungus. However, our fluorescence tests suggest a similar phenotype in MMV1593537-treated cryptococci and *C. auris*, pointing to the participation of chitinases in the effects induced by this compound.

Further experiments addressing acute toxicity *in vivo*, as well as the ability to control fungal diseases in mice, are required to take MMV1593537 to further steps of antifungal development. Nonetheless, our results suggest that MMV1593537 is a promising antifungal which might require chitinase activity to control fungal growth, at least in *Cryptococcus*.

## MATERIALS AND METHODS

**Compounds.** For the search of antifungal activity, we screened the MMV Pandemic Response Box compound library (Medicines for Malaria Venture, MMV, https://www.mmv.org/mmv-open/pandemic-response-box). The MMV Pandemic box contains 400 structurally diverse compounds which have already been marketed or are in various phases of drug discovery or development. These compounds are provided in 96-well plates at 10 mM in dimethyl sulfoxide. In our screen, we used standard isolates of *Cryptococcus neoformans* H99 and *C. deuterogattii* strain R265, and strains MMC1 and MMC2 of *Candida auris* (45). Antifungal activity tests with the most promising compound included previously characterized (46, 47) isolates of *Cryptococcus neoformans* (3Pb3, 17A1, 23Pb2, 19Pb4, Cg366, Cn161, Cn222, Cn116), *C. deuterogattii* (Cg460, Cg221, Cg158, Cg456, and Cg188), *C. gattii* (Cg365, Cg367, and Cg306) and *C. auris* (45) (CDC383, CDC388, CDC384, CDC390, CDC387, and CDC385). Isolates of *Candida albicans* (ATCC 90028 and ATCC MYA-2876), *C. parapsilosis* (ATCC 6258), and *C. krusei* (ATCC 22019) were obtained from the American Type Culture Collection. Stock cultures were stored in Sabouraud agar at 4°C. For antifungal tests, they were transferred to fresh Sabouraud agar, cultivated for 48 h at 30°C, and then used to inoculate the testing media.

**Antifungal screening.** The original 10 mM stock plates of the 400 compounds were first diluted in 100% DMSO to a concentration of 1 mM. Intermediate plates were prepared through a 1:20 (vol/vol) dilution in twice-concentrated (2×) RPMI medium (Sigma-Aldrich) supplemented with 2% glucose, buffered to pH 7.0 with 165 mM morpholinepropanesulfonic acid (MOPS). With this procedure, we generated plates containing the compounds at 50 $\mu$M in 5% DMSO. The plates used for the screening were prepared from a new dilution (1:2.5, vol/vol) of the intermediate plates in 2× RPMI medium supplemented with 2% glucose, generating plates with the compounds at 20 $\mu$M, 2% DMSO in a volume of 100 $\mu$L. After inoculation, they contained 10 $\mu$M of each compound in 1% DMSO. Following EUCAST antifungal susceptibility testing (AFST, E.DEF 7.3) protocol (21), *C. neoformans* H99, *C. deuterogattii* R265, *C. auris* MMC1, and *C. auris* MMC2 inocula were prepared in sterile water at a final density of 2.5 × 10⁵ cells/mL in 100 $\mu$L. These suspensions were used to inoculate the screening plates. All plates carried different controls: (i) sterility control (2× RPMI medium supplemented with 2% glucose [pH 7.0] with 165 mM MOPS and 2% DMSO, plus sterile water, no fungi); (ii) growth control (2× RPMI medium supplemented with 2% glucose [pH 7.0] with 165 mM MOPS and 2% DMSO, plus *C. neoformans* H99, *C. deuterogattii* R265, *C. auris* MMC1, or *C. auris* MMC2, no compounds); and (iii) antifungal activity control (amphotericin B at 0.5 mg/L). The plates were incubated at 35°C for 24 or 48 h for *C. auris* and *C. neoformans*, respectively. Antifungal activity was measured spectrophotometrically at 530 nm. The selection of compounds for further tests was based on their ability to preferentially inhibit at least 90% of the growth of the four fungal strains used in our screen. As detailed in Results, a few exceptions were allowed. The selected compounds were MMV1633966, MMV019724, MMV1593541, MMV565773, and MMV1593537. The Medicines for Malaria Venture initiative provided larger amounts of the compounds for the subsequent tests.

**Determination of the minimum inhibitory concentrations and minimum fungicidal concentrations of the selected compounds.** Determination of MICs was performed according to the EUCAST antifungal susceptibility testing (E.DEF 7.3) protocol (21). For this step, 96-well plates were prepared to contain the different compounds at concentrations ranging from 0.03 to 20 $\mu$M in 100 $\mu$L of 2× RPMI medium supplemented with 2% glucose and 165 mM MOPS (pH 7.0). The wells were inoculated with 100 $\mu$L of the fungal suspensions (2.5 × 10⁵ cells/mL) in sterile water. The plates were incubated at 35°C for 24 or 48 h for *C. auris* and *C. neoformans*, respectively. Antifungal activity was determined spectrophotometrically at 530 nm and the MIC corresponded to the smallest compound concentrations able to inhibit >90% of growth. To determine the MFC, 5 $\mu$L of each well from the MIC plates was transferred to Sabouraud agar plates and incubated at 30°C for 24 or 48 h for *C. auris* and *Cryptococcus* spp., respectively. The MFC was determined as the minimal concentration at which no fungal growth was observed.

**Cytotoxicity of the selected compounds.** The cytotoxicity of the selected compounds was determined using mouse macrophages as prototypes of host cells. RAW 264.7 macrophages (10⁵ cells) were cultivated in Dulbecco's Modified Eagle Medium (DMEM) supplemented with 10% fetal bovine serum and 0 to 10 $\mu$M of the selected compounds (100-$\mu$L suspensions distributed into the wells of 96-well plates). The plates were incubated for 24 h at 37°C with 5% $CO_2$. The supernatant was recovered and tested for lactate dehydrogenase activity using the Cytotox 96 Non-Radioactive Cytotoxicity kit (Promega) according to the manufacturer's recommendations. Control systems included vehicle (1% DMSO)-treated cells (viability control) and macrophage extracts using the lysis solution provided by the

manufacturer (death control). From this point on, we selected compound MMV1593537 for subsequent tests, due to its low toxicity to the cultured macrophages.

**Growth curves.** The effect of MMV1593537 on fungal growth was evaluated at three different concentrations: 1.25, 2.5, and 5 $\mu$M. DMSO was used as the vehicle and its concentration was maintained at 1% in all experiments. *C. neoformans* H99 and *C. deuterogattii* R265 were cultivated in the capsule induction medium (10% Sabouraud diluted in MOPS [pH 7.0]) (48) to allow the determination of direct antifungal effects and possible interference with the cryptococcal capsule. *C. auris* MMC1 and MMC2 were cultivated in Sabouraud broth. For all experiments, the four strains were pre-incubated overnight in Sabouraud broth at 30°C with agitation (150 rpm) and transferred to the cultivation medium for further incubation for 24 (*C. auris*) or 48 (cryptococci) h at 37°C. For the growth curves, an inoculum of $2.5 \times 10^5$ cells/mL of each strain was incubated with the different MMV1593537 concentrations at 37°C. Fungal growth was determined on a Molecular Devices SpectraMax Paradigm microplate reader, with optical density measurements at 530 nm every hour.

**Preparation of fungal cells for the analysis of fungal morphology.** For analysis of fungal morphology, *C. neoformans* H99, *C. deuterogattii* R265, *C. auris* MMC1, and *C. auris* MMC2 were pre-incubated overnight in Sabouraud broth at 30°C with agitation (150 rpm), washed three times with phosphate-buffered saline (PBS), and the suspensions were adjusted to $2.5 \times 10^5$ cells/mL in the capsule induction medium (*Cryptococcus* spp) or liquid Sabouraud (*C. auris*). These cell suspensions contained either MMV1593537 at 2.5 $\mu$M plus 1% DMSO or only 1% DMSO as a control. The cells were incubated at 37°C under a 5% $CO_2$ atmosphere for 24 h. The cultures were washed 3 times and processed for different microscopic approaches, described as follows.

**Scanning electron microscopy analysis.** Control or MMV1593537-treated cells were washed 3 times with PBS and fixed with 2.5% glutaraldehyde in 0.1 M sodium cacodylate buffer (pH 7.2) for 1 h at room temperature. The cells were washed 3 times with 0.1 M sodium cacodylate buffer (pH 7.2) containing 0.2 M sucrose and 2 mM $MgCl_2$ and placed over 0.01% poly-L-lysine-coated coverslips. The fixed cells were allowed adhere to the coverslips for 1 h at room temperature, followed by dehydration in ethanol (30, 50, and 70% for 5 min, 90% for 10 min, and 100% twice for 10 min). Dehydrated cells were critical point-dried (Leica EM CPD300), mounted on metallic bases, and coated with a gold layer (Leica EM ACE200). The cells were visualized using a scanning electron microscope (JEOL JSM-6010 Plus/LA) operating at 10 keV.

**Light microscopy.** Control or MMV1593537-treated cryptococci were washed 3 times with PBS, fixed with 4% paraformaldehyde, and microscopically observed for quantitative analysis of the capsule. Fixed cells were counterstained with India Ink [5] and observed under a DMi8 microscope (Leica). Images were recorded using the LasAF software (Leica), and cell body and capsule dimensions were determined in digitalized images using the ImageJ software (49).

**Fluorescence microscopy.** Control or MMV1593537-treated *Cryptococcus* spp. and *C. auris* were fixed with 4% paraformaldehyde and washed 3 times with PBS. The fixed cells were blocked with 1% bovine serum albumin (BSA) in PBS for 1 h at 37°C, following chitin staining at the cell wall with 25 $\mu$M Calcofluor white (Sigma-Aldrich) for 30 min at 37°C. The cells were washed 3 times with PBS and incubated with 5 $\mu$g/mL wheat germ agglutinin-tetramethylrhodamine conjugate in PBS for 30 min at 37°C. After this incubation, the cells were washed 3 times and analyzed under a DMi8 microscope (Leica). Images were recorded with LasAF software (Leica) and processed with ImageJ (49).

**Chitinase activity.** Chitinase activity was measured in fungal cells that were exposed to MMV1593537. Fungal cells were cultivated overnight in liquid Sabouraud at 30°C, then washed with PBS and adjusted to a cell density of $10^6$ cells/mL in capsule induction medium (cryptococci) or Sabouraud broth (*C. auris*) supplemented with a subinhibitory concentration of either MMV1593537 (2.5 $\mu$M) plus 1% DMSO or only 1% DMSO as the control. The cultures were incubated at 37°C for 0, 20, 40, 60, 120, 240, 360, and 1,440 min. The cells were then centrifuged and washed three times with PBS. For measurement of chitinase activity, $10^6$ cells were suspended in 180 $\mu$L of 6 mM 4-methylumbelliferyl *N*-acetyl-$\beta$-D-glucosaminide (chitinase substrate, Sigma-Aldrich) in phosphate-citrate buffer (40 mM sodium citrate, 88 mM sodium phosphate dibasic [pH 4.5]) and incubated for 1 h at 37°C. The cells were removed by centrifugation, the supernatants (160 $\mu$L) were transferred to a new tube, and the reaction was stopped with 40 $\mu$L of 0.1 M glycine buffer (pH 10). The enzyme activity was measured on a fluorimeter (wavelength: 360 to 450 nm, Synergy H1 Hybrid Multi-Mode Reader, BioTek). Measurement of enzyme activity included monitoring of cell viability. The cells were prepared and incubated as described above, and at the 0, 2, and 24 h time points, aliquots of 1 mL were taken. The cells were washed 3 times with PBS and resuspended in 1 mL PBS. Each aliquot was diluted in PBS (1:10, 1:100, and 1:1,000), and 100 $\mu$L of each dilution was inoculated on Sabouraud agar plates. After incubation for at 30°C for 48 h, CFU/mL was determined.

**Statistical analysis.** Statistical analyses were performed using Graph Pad Prism (GraphPad Software, San Diego, CA, USA), and results were considered significant when *P* values of $<0.05$ were obtained. In the capsule size assay, the results were analyzed using one-way analysis of variance and Tukey's *post hoc* tests.

## ACKNOWLEDGMENTS

We thank the Medicines for Malaria Venture for providing the Pandemic Response Box. M.L.R. is supported by grants from the Brazilian Ministry of Health (grant 440015/2018-9), Cons-elho Nacional de Desenvolvimento Científico e Tecnológico (grants 405520/2018-2 and 301304/2017-3), and Fiocruz (grants PROEP-ICC 442186/2019-3, VPPCB-007-FIO-18, and VPPIS-001-FIO18). M.L.R. also acknowledges support from the

Instituto Nacional de Ciência e Tecnologia de Inovação em Doenças de Populações Negligenciadas. H.C.dO. received scholarships from the Inova Program of Fiocruz. F.C.G.R. received a scholarship from the Coordenação de Aperfeiçoamento de Pessoal de Nível Superior (Brazil, Finance Code 001). J.D.N. is supported in part by NIH R21AI156104.

The funders had no role in the decision to publish, or preparation of the manuscript.

We are grateful to the Program for Technological Development in Tools for Health (RPT-FIOCRUZ) for use of the microscopy facility (RPT07C, Carlos Chagas Institute, Fiocruz, Paraná). M.L.R. is currently on leave from the position of associate professor at the Microbiology Institute of the Federal University of Rio de Janeiro, Brazil.

We have no conflicts of interest to report.

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
