## [Reviewer comments · Microbiology Spectrum]

Microbiology Spectrum

Screening of the Pandemic Response Box reveals an association between antifungal effects of MMV1593537 and the cell wall of *Cryptococcus neoformans*, *Cryptococcus deuterogattii*, and *Candida auris*

Haroldo de Oliveira, Rafael Castelli, Flavia Reis, Kirandeep Samby, Joshua Nosanchuck, Lysangela Alves, and Marcio Rodrigues

Corresponding Author(s): Marcio Rodrigues, Fundação Oswaldo Cruz-Fiocruz

Review Timeline:

Submission Date:	February 16, 2022
Editorial Decision:	March 19, 2022
Revision Received:	March 24, 2022
Accepted:	April 11, 2022

Editor: Kirsten Nielsen

Reviewer(s): The reviewers have opted to remain anonymous.

Transaction Report:

DOI: <https://doi.org/10.1128/spectrum.00601-22>

March 19, 2022

Dr. Marcio L. Rodrigues
Fundação Oswaldo Cruz-Fiocruz
Carlos Chagas Institute
Curitiba, Paraná
Brazil

Re: Spectrum00601-22 (Screening of the Pandemic Response Box reveals an association between antifungal effects of MMV1593537 and the cell wall of *Cryptococcus neoformans*, *Cryptococcus deuterogattii*, and *Candida auris*)

Dear Dr. Marcio L. Rodrigues:

The reviewers highlight only very minor modifications to the manuscript. Please respond to the reviewer comments with appropriate edits to the manuscript.

Link Not Available

Sincerely,

Kirsten Nielsen

Journals Department
Reviewer comments:

Reviewer #1 (Comments for the Author):

Summary of the manuscript:

The manuscript tested the antifungal activity of 400 compounds from the pandemic response box against fungal pathogens *Cryptococcus neoformans*, *Cryptococcus deuterogattii* and *Candida auris*. Initial screening identified 5 compounds that inhibited the growth of the three fungal pathogens at the concentration of 10 μ M. The MIC of the identified 5 compounds were measured as well as their toxicity to mammalian cells (RAW macrophages). Four compounds were found to be toxic to mammalian cells or

cause the fungal paradoxical antifungal effects, and were eliminated from the screening. The one compound left (MMV1593537) had high fungicidal effect and low toxicity to macrophages, and was selected for further characterisation of its effect on the three fungal pathogens. This analysis found that the selected compound (MMV1593537) affected cellular divisions in all three fungi tested, and increased chitooligomers in both *Cryptococcus* strains and one strain of *Candida auris*. In *Cryptococcus* MMV1593537 induced changes in capsule structure, a reduction in the capsule sizes and increased chitinase activity.

Comments:

This is a well written manuscript. The study described in the manuscript aimed to contribute to addressing the challenge of lack of effective antifungal drugs against the fungal pathogens *Cryptococcus* species and *Candida auris*. The methods used are clear and appropriate. The conclusions made by the authors are supported by the data presented. However, some areas were not clear. Below are my questions on the areas that are not clear and some suggestions.

Questions and suggestions

1. Materials & Methods:

- a. Line 115 or in Table 3: I would suggest adding the references of where different isolates have been characterised.
- b. Line 120: how long were the stock cultures were maintained at 30{degree sign}C?
- c. Line 240: for testing the chitinase activity 2.5 μM of MMV1593537 was used. Why using this specific concentration?

2. Figure 1 and Figure 2: The compound MMV1633966 inhibited at least 90% of the growth of *Cryptococcus* using the concentration of 10 μM (indicated on lines 131-132). However, figure 2 shows *Cryptococcus* growth of 50-70% with the same compound at 10 μM . Why the discrepancy if the same conditions (media, compound concentration, and temperature) were used?

3. Figure 2: does the Y-axis represent 'percentage of growth inhibition' or just 'percentage' of growth? If it is the growth inhibition as written in the current manuscript, then no compound (0 μM) is inhibiting almost 100% of fungal growth while the highest compound concentration is inhibiting zero to around 10% of the growth. This looks like a mistake that Y-axis is actually showing the percentage of growth.

4. Figure 7: On images showing *Cryptococcus* cells, it is good to see bigger images to observe the capsule structure. However these bigger images only show few cells. I would have liked to also see large scale images showing many cells (similar to what was done in Fig 6) for the purpose of observing and comparing the formation of cell aggregates between *Candida auris* and *Cryptococcus* (to show what is written in lines 346-348).

5. Figure 7: I would suggest adding the statistical test used to compare capsule sizes to the figure legend.

6. Figure 8 and lines 365-366, it looks like there is more CFW staining of *Cryptococcus* R265 after MMV1593537 treatment. In the analysis to conclude that there is no difference in chitin staining, were the fluorescence intensities compared or just the visual observation?

Reviewer #2 (Comments for the Author):

In the article titled "Screening of the Pandemic Response Box reveals an association between antifungal effects of MMV1593537 and the cell wall of *Cryptococcus neoformans*, *Cryptococcus deuterogattii*, and *Candida auris*", Oliveira et. al identify several antifungal molecules from the Pandemic Response Box. Five compounds exhibited anti-cryptococcal and anti-candida auris activity. Two of these compounds (MMV1593537 and MMV1633966) did not exhibit toxicity against mouse macrophages at or below their respective MICs. MMV1593537 was selected as the most promising compound for further studies as MMV1633966 did not exhibit fungicidal activity and exhibited a paradoxical effect similar to the echinocandins. MMV1593537 exhibited activity across diverse *Cryptococcus* and *Candida* species. Sub-inhibitory concentrations of this compound induced aggregates in *Candida* and reduced capsule size in *Cryptococcus*. This observation led to the hypothesis that MMV1593537 induced a cell wall defect. While no difference in chitin (Calcofluor white) staining was observed with MMV1593537 treatment, a stark increase and wider distribution in chitooligomer (WGA) staining was observed. An increase in chitinase activity was detectable in *Cryptococcus* species but not *Candida*. Together, the authors present a nice reference for anti-cryptococcal and anti-candida hits from the Pandemic Response Box and provide a good lead on one of these compounds for further development.

Major Points - None

Minor Points

Figure 2: y-axis, % growth rather than % growth inhibition, a color coded key for the strains on the figure would be helpful rather than referring to legend. Also, please log transform the x-axis to better visualize lower concentrations.

Chitinase Assay, please identify and show a positive control. How does this chitinase activity signal compare to another strain or condition known to induce chitinase activity? Why does the chitinase activity decline? Does this correspond with cell death? You might consider performing a Time-kill.

The lack of chitinase activity in candida species might indicate perhaps chitinase is not driving the increase in chitooligomers.

The structure of MMV1633966 in Table 1 appears with additional lines.

Staff Comments:

Preparing Revision Guidelines

Please return the manuscript within 60 days; if you cannot complete the modification within this time period, please contact me. If you do not wish to modify the manuscript and prefer to submit it to another journal, please notify me of your decision immediately so that the manuscript may be formally withdrawn from consideration by Microbiology Spectrum.

Reviewer 1

Questions and suggestions

1. Materials & Methods:

a. Line 115 or in Table 3: I would suggest adding the references of where different isolates have been characterised.

Authors' response and action taken during revision: We agree with the reviewer. We added the appropriate references and, for the two *Candida albicans* isolates, we adopted the standard ATCC nomenclature. Please see lines 113 – 119 and Table 3 in the revised manuscript.

b. Line 120: how long were the stock cultures were maintained at 30{degree sign}C?

Authors' response and action taken during revision: Thank you for this question, which guided us to a more complete description of our stock cultures. As detailed in the revised manuscript (lines 199-121), stock cultures were stored in Sabouraud agar at 4°C. For the antifungal tests, they were transferred to fresh Sabouraud agar, cultivated for 48 h at 30°C, and then used to inoculate the testing media.

c. Line 240: for testing the chitinase activity 2.5 µM of MMV1593537 was used. Why using this specific concentration?

Authors' response and action taken during revision: To make sure the cells were viable we used a sub-inhibitory concentration of MMV1593537. Please see lines 240-241 of the revised manuscript.

2. Figure 1 and Figure 2: The compound MMV1633966 inhibited at least 90% of the growth of *Cryptococcus* using the concentration of 10 µM (indicated on lines 131-132). However, figure 2 shows *Cryptococcus* growth of 50-70% with the same compound at 10 µM. Why the discrepancy if the same conditions (media, compound concentration, and temperature) were used?

Authors' response and action taken during revision: Nice point. One of the reasons by which MMV1633966 was left behind in our study was its inconsistency in inhibiting fungal growth at higher concentrations. This view is now detailed in the revised manuscript (lines 291-294), where we explain that in the dose-response tests, the percentage of growth inhibition at 10 µM was repeatedly lower than that observed in the full screenings, in addition to the paradoxical-like effect observed for this compound.

3. Figure 2: does the Y-axis represent 'percentage of growth inhibition' or just 'percentage' of growth? If it is the growth inhibition as written in the current manuscript, then no compound (0 µM) is inhibiting almost 100% of fungal growth while the highest compound concentration is inhibiting zero to around 10% of the growth. This looks like a mistake that Y-axis is actually showing the percentage of growth.

Authors' response and action taken during revision: Thank you very much for the careful analysis of our manuscript. Indeed, Y-axis refers to the percentage of growth. The figure was modified to eliminate this mistake. Another modification was the presentation of the X-axis in a log scale, as required by Reviewer 2.

4. Figure 7: On images showing *Cryptococcus* cells, it is good to see bigger images to observe the capsule structure. However these bigger images only show few cells. I would have liked to also see large scale images showing many cells (similar to what was done in Fig 6) for the purpose of observing and comparing the formation of cell aggregates between *Candida. auris*

and *Cryptococcus* (to show what is written in lines 346-348).

Authors' response and action taken during revision: We agree with the reviewer and modified the Figure to include additional microscopic fields with more cells. *C. neoformans* H99, but not *C. deuterogattii* R265, formed discrete aggregates, but they are not comparable to those formed by *C. auris* cells. This view is now detailed in the manuscript (line 354 and legend for Figure 7).

5. Figure 7: I would suggest adding the statistical test used to compare capsule sizes to the figure legend.

Authors' response and action taken during revision: Done.

6. Figure 8 and lines 371-377, it looks like there is more CFW staining of *Cryptococcus* R265 after MMV1593537 treatment. In the analysis to conclude that there is no difference in chitin staining, were the fluorescence intensities compared or just the visual observation?

Authors' response and action taken during revision: Nice point. We used our figures to measure the intensity of calcofluor-derived fluorescence. There was a discrete increase in MMV1593537-treated *C. auris* (strain MMC2 only). Please see lines 365-371 of the revised manuscript.

Reviewer 2

Minor Points

Figure 2: y-axis, % growth rather than % growth inhibition, a color coded key for the strains on the figure would be helpful rather than referring to legend. Also, please log transform the x-axis to better visualize lower concentrations.

Authors' response and action taken during revision: As well pointed-out by reviewers 1 and 2, there was a mistake with what we originally called % of growth inhibition. This mistake was corrected and, as required by Reviewer 2, we changed the x-axis to a log scale. Indeed, the figure is more informative now; thanks for the suggestions.

Chitinase Assay, please identify and show a positive control. How does this chitinase activity signal compare to another strain or condition known to induce chitinase activity? Why does the chitinase activity decline? Does this correspond with cell death? You might consider performing a Time-kill.

Authors' response and action taken during revision: The reviewer raised important points. As for the positive control, we considered using purified enzymes that are commercially or even conditions that we used previously to explore the connections between chitin synthesis and chitinase in *C. neoformans* (<https://doi.org/10.1016/j.tcsw.2018.05.002>). However, none of these conditions were minimally comparable to our current experimental models, so we decided to not include it to avoid the artificialization of our conclusions. As for the conditions known to induce chitinase activity, we knew from this same previous study that stressing conditions – like those used for capsule growth – induced chitinase activity in *C. neoformans* (lines 456-457 of the revised manuscript), and we used the same rationale. As for the decline in enzyme activity, we checked cell death under these conditions, as suggested by the reviewer. We determined CFUs at the 0, 2, and 24 h time points for *C. neoformans* and *C. deuterogattii* R265 and observed that fungal viability increased with time in both control and MMV1593537-treated cells. This information was included in the revised manuscript – please see lines 250-256 (Methods), 399-404 (Results), and 755-759 (legend for Figure 9).

The lack of chitinase activity in candida species might indicate perhaps chitinase is not driving the increase in chitooligomers.

Authors' response and action taken during revision: Correct. Please see lines 469-471 of the revised manuscript, where we included this view.

The structure of MMV1633966 in Table 1 appears with additional lines.

Authors' response and action taken during revision: Correct. We changed the structure to its conventional form showing the double bonds in the cyclic rings.

April 11, 2022

Dr. Marcio L. Rodrigues
Fundação Oswaldo Cruz-Fiocruz
Carlos Chagas Institute
Curitiba, Paraná
Brazil

Re: Spectrum00601-22R1 (Screening of the Pandemic Response Box reveals an association between antifungal effects of MMV1593537 and the cell wall of *Cryptococcus neoformans*, *Cryptococcus deuterogattii*, and *Candida auris*)

Dear Dr. Marcio L. Rodrigues:

Your manuscript has been accepted, and I am forwarding it to the ASM Journals Department for publication. You will be notified when your proofs are ready to be viewed.

Sincerely,

Kirsten Nielsen
Editor, Microbiology Spectrum
